# Identification of Potential Druggable Targets and Structure-Based Virtual Screening for Drug-like Molecules against the Shrimp Pathogen *Enterocytozoon hepatopenaei*

**DOI:** 10.3390/ijms24021412

**Published:** 2023-01-11

**Authors:** Prasenjit Paria, Anchalee Tassanakajon

**Affiliations:** Center of Excellence for Molecular Biology and Genomics of Shrimp, Department of Biochemistry, Faculty of Science, Chulalongkorn University, Bangkok 10330, Thailand

**Keywords:** *Enterocytozoon hepatopenaei*, drug molecule, virtual screening, docking, molecular dynamics simulation

## Abstract

*Enterocytozoon hepatopenaei* (EHP) causes slow growth syndrome in shrimp, resulting in huge economic losses for the global shrimp industry. Despite worldwide reports, there are no effective therapeutics for controlling EHP infections. In this study, five potential druggable targets of EHP, namely, aquaporin (AQP), cytidine triphosphate (CTP) synthase, thymidine kinase (TK), methionine aminopeptidase2 (MetAP2), and dihydrofolate reductase (DHFR), were identified via functional classification of the whole EHP proteome. The three-dimensional structures of the proteins were constructed using the artificial-intelligence-based program AlphaFold 2. Following the prediction of druggable sites, the ZINC15 and ChEMBL databases were screened against targets using docking-based virtual screening. Molecules with affinity scores ≥ 7.5 and numbers of interactions ≥ 9 were initially selected and subsequently enriched based on their ADMET properties and electrostatic complementarities. Five compounds were finally selected against each target based on their complex stabilities and binding energies. The compounds CHEMBL3703838, CHEMBL2132563, and CHEMBL133039 were selected against AQP; CHEMBL1091856, CHEMBL1162979, and CHEMBL525202 against CTP synthase; CHEMBL4078273, CHEMBL1683320, and CHEMBL3674540 against TK; CHEMBL340488, CHEMBL1966988, and ZINC000828645375 against DHFR; and CHEMBL3913373, ZINC000016682972, and CHEMBL3142997 against MetAP2.The compounds exhibited high stabilities and low binding free energies, indicating their abilities to suppress EHP infections; however, further validation is necessary for determining their efficacy.

## 1. Introduction

*Enterocytozoon hepatopenaei* (EHP) infections in Pacific white shrimp, *Litopenaeusvannamei*, have become a pandemic in recent years [1]. The pathogen was first discovered and isolated from the hepatopancreatic tissues of black tiger shrimp, *Penaeus monodon*, in Thailand in 2009 [2]. EHP infections were subsequently reported in China, India, Vietnam, Indonesia, Malaysia, Venezuela, and Australasia [1,3,4]. EHP is an intracellular spore-forming parasite that germinates within the epithelial cells of the hepatopancreas and is transmitted horizontally [1,5]. The presence of EHP spores in the hepatopancreas and midgut indicates an oral route of transmission, which could either result from immersion, swallowing pathogen-infected water, active cannibalism, or the consumption of infected shrimp [3,5]. A recent analysis of the 3.26 Mbp EHP genome revealed that the pathogen could not generate ATP either viaglycolysis or oxidative phosphorylation [6,7]. EHP, therefore, obtains ATP and other purine nucleotides for energy and biosynthesis from their host by using nucleotide transport (NTT) proteins, which are acquired via lateral gene transfer [8].

Although EHP infections do not cause significant mortality in shrimp, the pathogen dramatically retards the growth of penaeid shrimp and is therefore emerging as a critical threat to the shrimp farming industry, owing to the loss of more than 7.4 million USD annually [9]. Significant research efforts have been made to understand the mechanisms of EHP transmission and pathogenicity, the detection of EHP infections, and the identification of EHP-resistant or -tolerant shrimp germlines for reducing disease outbreaks in the aquaculture industry. Because the parasite is spreading to new geographical locations, practical control measures are being applied to prevent infections from environmental sources. However, it is extremely difficult to mitigate the situation once EHP infiltrates the system, owing to the lack of effective or approved therapeutic methods at present. Virtual screening (VS) is a powerful computational approach that involves high throughput screening (HTS) for screening large libraries of drug molecules against selected active biological macromolecules [10]. Structure-based VS (SBVS) is widely used for screening novel drug molecules against various parasites, including the malarial parasite [11], *Toxoplasma gondii* [12], *Trypanosoma brucei* [13], and *Leishmania* sp. [14]. Therefore, the present study aimed to identify novel compounds with high binding affinities and high chemical complementarities with the potential druggable target proteins of EHP by applying VS, molecular docking, and molecular dynamics (MD) simulations.

## 2. Results

### 2.1. Identification of Potential Druggable Target Proteins and Generation of Three-Dimensional Models

A total of five potential druggable target proteins of EHP were identified, of which three proteins, namely, cytidine triphosphate (CTP) synthase, thymidine kinase (TK), and dihydrofolate reductase (DHFR), are associated with the biosyntheses of pyrimidine nucleotides and purine nucleotides. Of the other two proteins, methionine aminopeptidase (MetAP) is an exopeptidase, while aquaporin (AQP) is responsible for the maintenance of cellular osmotic pressure. The sequences of these 5proteins were compared with the proteome data of *L. vannamei* available in the UniProt Knowledgebase (https://www.uniprot.org/ accessed on 5 January 2022), and the results demonstrated that all the selected proteins have <25% identity with the proteins of the host. Therefore, the compounds identified against the five proteins of EHP are less likely to inhibit the proteins of the host.

The three-dimensional structures of all the five proteins of EHP were generated using AlphaFold 2 (Figure 1). The structures were further validated with different protein structure validation tools for determining the reliabilities of the modeled structures. Analysis of the Ramachandran plot revealed that 99% of the amino acid residues of all the predicted structures were in the allowed regions, while only 0.7% of the residues were in the disallowed regions of the plot. The ProSA server was used to estimate the overall model qualities, indicated by the value of the ProSA Z-score. The ProSA Z-scores of the models of AQP (Appendix A), CTP synthase (Appendix A), TK (Appendix A), DHFR (Appendix A), and MetAP (Appendix A) were −11.22, −5.88, −7.56, −8.94, and −3.77, respectively, which indicated that the qualities of the models were similar to those of the protein structures predicted using NMR and X-ray crystallography.

### 2.2. Identification of Pockets Based on Conserved Amino Acid Residues in Active Sites

The sequence homologs of the query proteins were determined with BLASTp against the Protein Data Bank (PDB). EHP AQP (A0A1W0E445) had 95% query coverage and 29% identity with the AQP protein of *Arabidopsis thaliana* (PDB ID: 5I32), and 98% query coverage and 26% identity with human AQP (PDB ID: 1FQY). Residues Phe22, Asn69, Pro70, Glu138, Gly197, and Asn201 were found to be conserved in EHP AQP (Appendix A). Similarly, the CTP synthase of EHP (A0A1W0E736) showed 97% and 96% query coverage with the CTP synthase proteins of *Drosophila melanogaster* (PDB ID: 6L6Z) and human (PDB ID: 5U03), respectively, and 29% and 27% identity, respectively. Residues Lys24, Tyr295, Cys377, Phe351, and Arg453 were conserved in the CTP synthase of EHP (Appendix A). The DHFR protein of EHP (A0A1W0E362) showed 63% and 64% query coverage with the DHFR of *Coxiella burnetii* (PDB ID: 3TQ8) and human (PDB ID: 3F8Y), respectively, and 30% and 28% identity, respectively. Furthermore, residues Leu27, Ala29, Met43, Trp45, Asp51, Lys56, Arg71, and Thr138 were found to be conserved in the DHFR protein of EHP (Appendix A). The MetAP2 protein (A0A1W0E5M9) of EHP showed 99% and 95% query coverage with the MetAP2 proteins of *Encephalitozoon cuniculi* (PDB ID: 3FM3) and human (PDB ID: 1R58), respectively, and 51% and 43% sequence identity, respectively. Residues Phe80, Pro81, His91, Ile201, His202, His244, Met246, Pro275, and Tyr304 were conserved in the MetAP2 protein of EHP (Appendix A). The binding pockets of all the druggable targets were predicted using the CASTp server, and the potential druggable pockets were selected based on the conserved amino acid residues (Table 1). No significant homologs of EHP TK (A0A1W0E7R0) were identified in the PDB. A sequence-based comparison among the host proteins and EHP proteins was carried out to identify the total amino acids present in the binding pockets of host proteins. The detailed comparison study is mentioned in Table 2.

### 2.3. Docking-Based Virtual Screening

The ZINC Purchase library and the ChEMBL database, comprising 12,941,912 and 1,791,953 drug-like molecules, respectively, were screened against the predicted ligand-binding pockets of the 5 druggable targets of EHP. After the primary screening, approximately 1100 drug molecules were selected against each protein target from the ZINC15 library and ChEMBL database based on Lipinski’s rule of five. The selected molecules were further screened using docking-based VS. Compounds with affinity scores ≥ −7.5 and ≥ 9.0 protein-ligand interactions were selected. Totals of 247, 183, 198, 98, and 128 compounds were finally selected against CTP synthase, TK, MetAP2, DHFR, and AQP, respectively (Appendix A).

### 2.4. Prediction of ADMET Properties

The ADMET properties of the screened drug-like molecules were subsequently predicted. The results demonstrated that most of the compounds qualified the Golden Triangle rule, Lipinski’s rule of five, and Pfizer’s rule. Molecules with predicted carcinogenicity, rat oral acute toxicity, AMES toxicity, and mutagenicity potentials were removed from the library prior to the final screening. Totals of 156, 120, 115, 73, and 81 compounds were finally selected against CTP synthase, TK, MetAP2, DHFR, and AQP, respectively (Appendix A). The selected compounds were further screened based on their electrostatic complementarity (EC) scores.

### 2.5. EC-Based Screening

The selected molecules were further screened based on their EC using Flare v5.0.0, and compounds with EC scores > 0.25 were selected. For each protein, the top 20 ligands were generally selected from the docked conformations for further analysis (Appendix A). The PatchDock scores of the selected compounds were additionally determined. The five best compounds were finally selected based on their affinity scores, EC scores, PatchDock scores, and the numbers of hydrogen bonds (Appendix A). The ADMET properties of the five best compounds are provided in Appendix A.

### 2.6. Analysis of Protein-Ligand Interactions

The intermolecular interactions between the selected compounds and the binding sites of the target proteins are depicted in Figure 2. The results demonstrated that hydrogen bonds, van der Waals interactions, and carbon–hydrogen bonds played significant roles in maintaining complex stability. The intermolecular interactions were analyzed from the best docked pose for each molecule. Analysis of the ligand interactions of AQP revealed that all the five compounds, namely, CHEMBL3703838, ZINC000002243083, CHEMBL133039, CHEMBL3140193, and CHEMBL2132563, were stabilized via hydrogen bonds with different amino acids atthe binding site (Table 3). The calculated hydrogen bond distances ranged from 2.3 to 2.9 Å. ZINC000002243083 and CHEMBL3140193 formed the highest numbers of hydrogen bonds with the binding site. Further analysis revealed that three compounds, CHEMBL3703838, CHEMBL133039, and CHEMBL3140193, formed π–π stacking interactions with Thr121, Lys34, and Thr33, respectively (Figure 2A–E). The compounds were stabilized by the surrounding residues via various protein-ligand interactions, including hydrogen bonds, van der Waals interactions, and electrostatic interactions. The binding affinities of the 5 compounds (CHEMBL48494, CHEMBL1162979, CHEMBL133039, CHEMBL525202, and CHEMBL1091856) selected against CTP synthase ranged from−9.6 to −7.6 kcal/mol (Table 3). Analysis of the protein-ligand interactions revealed that all the compounds were stabilized in the binding pocket of CTP synthase via hydrogen bonds. The calculated hydrogen bond distances ranged from 1.7 to 2.9 Å. CHEMBL1091856 formed the highest number of hydrogen bonds, followed by CHEMBL1162979 and CHEMBL525202. With the exception ofCHEMBL525202, all the compounds formed π–π stacking interactions with the receptors. CHEMBL48494, CHEMBL1091856, and ZINC000219968783 formed π–π interactions with Phe351, while CHEMBL133039 formed three π–π interactions with Phe351, Arg453, and Tyr302 (Figure 2F–J).

A total of 5 compounds, CHEMBL3674540, CHEMBL1683320, CHEMBL391279, ZINC000031750813, and CHEMBL4078273, were similarly selected against TK, with binding affinities ranging from −9.5 to −8.1 kcal/mol (Table 3). All the compounds were stabilized in the binding pocket via hydrogen bonds, and the calculated hydrogen bond distances ranged from 1.8 to 3.0 Å. CHEMBL1683320 formed the highest number of hydrogen bonds, followed by CHEMBL391279 and ZINC000031750813. Analysis of the docked poses of compounds CHEMBL391279, ZINC000031750813, and CHEMBL4078273 revealed additional π–π stacking interactions with Arg45 and Arg46 (Figure 2K–O).

A total of five compounds, namely, CHEMBL56533, ZINC000016682862, ZINC000828645375, CHEMBL3901573, and CHEMBL108166, were selected against EHP DHFR. All the compounds formed hydrogen bonds with different amino acids and were stabilized in the binding pocket (Table 3). The calculated hydrogen bond distances ranged from 1.7 to 3.0 Å. CHEMBL3901573, ZINC000828645375, CHEMBL108166, and ZINC000016682862 formed additional π–π stacking interactions with different amino acids. CHEMBL3901573 and CHEMBL108166 formed π–π interactions with Met55 and Arg150, respectively. ZINC000828645375 formed three π–π interactions with Met55, Phe52, and Tyr79. ZINC000016682862 formed two π–π stacking interactions with Tyr79 and Phe52. CHEMBL56533 formed the highest number of hydrogen bonds, followed by CHEMBL108166 (Figure 2P–O). Similarly, a total of five compounds, namely, CHEMBL3913373, CHEMBL1962731, CHEMBL3142997, ZINC000199197855, and ZINC000016682972, were selected against MetAP2 using docking-based VS (Table 3). All the compounds formed hydrogen bonds with different amino acids and were stabilized in the binding pocket of the receptor. The calculated hydrogen bond distances ranged from 1.8 to 2.9 Å. CHEMBL3913373 and ZINC000199197855 formed the highest number of hydrogen bonds. CHEMBL3913373, CHEMBL1962731, ZINC000199197855, and ZINC000016682972 formed π–π interactions with different amino acids in the binding pocket. CHEMBL3913373 formed a single π–π stacking interaction with His92. However, CHEMBL1962731, ZINC000199197855, and ZINC000016682972 formed two π–π interactions with His92 as well as His202, Tyr304, and Phe80, respectively (Figure 2U–Y).

### 2.7. MD Simulations of Protein-Ligand Complexes

MD is a powerful computational method for predicting and analyzing the stabilities of protein-ligand complexes and for studying atomic movements with respect to a macromolecule. The stabilities and behaviors of the protein-ligand complexes were analyzed in a dynamic environment based onroot-mean-square deviation (RMSD), root-mean-square fluctuation (RMSF), the radius of gyration (Rg), and molecular mechanics/generalized Born surface area (MM/GBSA) energy. The simulation trajectories were subjected to a principal component analysis (PCA), and a cross-correlation matrix of the resultant MD trajectories was also constructed and analyzed.

#### 2.7.1. RMSD Values of the Cα Backbone of Target Proteins

The RMSD values of the protein-ligand complexes were plotted graphically for understanding the structural stability and integrity of the complexes. The results demonstrated that the average RMSD values of the 5 compounds selected against AQP ranged from 1.84 to 3.68 Å. Compounds CHEMBL3703838, ZINC000002243083, and CHEMBL3140193 showed high stabilities during the simulation, with an RMSD fluctuation of 0.5 Å (Figure 3A, Appendix A). Similarly, CHEMBL48494, CHEMBL133039, CHEMBL525202, and CHEMBL1091856 formed stable complexes with CTP synthase, with the average RMSD values ranging from 1.90 to 2.49 Å (Figure 4A, Appendix A). The average RMSD values for the Cα backbone of TK complexed with the 5 compounds ranged from 2.13 to 6.96 Å (Figure 5A, Appendix A). The average RMSD value of ZINC000031750813 (6.96 Å) was higher than that of the four other compounds selected against TK (Appendix A). With the exceptions of CHEMBL3901573 and ZINC000016682862, the average RMSD values of the three other compounds selected against DHFR indicated stable bindings (Figure 6A, Appendix A). The average RMSD fluctuation of these compounds ranged from 2.97 to 4.22 Å. The trajectory analysis revealed that all the 5 compounds complexed with MetAP2 remained stable throughout the 100 ns simulation, with the average RMSD values of the compounds ranging from 1.21 to 1.37 Å (Figure 7A, Appendix A).

#### 2.7.2. RgValues of the Cα Backbone of Target Proteins

The compactness of the protein-ligand complexes during the simulation was determined by measuring the values of Rg. The average values of Rg for the 5 compounds complexed with AQP ranged from 17.89 to 18.30 Å (Figure 3B, Appendix A). Similarly, the average Rg values of the compounds complexed with CTP synthase ranged from 25.11 to 25.42 Å (Figure 4B, Appendix A). With the exception of ZINC000031750813 (19.66 Å), all the 4 compounds selected against TK formed stable complexes with the receptor, with the average Rg values ranging from 17.89 to 18.05 Å (Figure 5B, Appendix A). The average Rg values of the 5 compounds selected against DHFR ranged from 16.98 to 17.48 Å. With the exceptions of ZINC000016682862 and CHEMBL3901573, the three compounds selected against DHFR formed stable complexes with the receptor (Figure 6B, Appendix A). All five of the compounds complexed with MetAP2 formed tightly packed, stable structures with the receptor (Figure 7B, Appendix A).

#### 2.7.3. RMSF Values of the Cα Backbone of Target Proteins

The average atomic mobility of the protein backbone during the MD simulations was measured using the values of RMSF. The average RMSF values of the 5 compounds complexed with AQP ranged from 0.92 to 1.53 Å (Figure 3C, Appendix A). Further analysis revealed that residues 160–170 of AQP underwent fluctuations when complexed with CHEMBL3703838, CHEMBL133039, and CHEMBL2132563. Residues Thr33, Lys34, Gly46, Leu119, Ile120, Ala123, Ser193, Ser196, Gly197, Gly198, and Ala 199 of AQP, which were primarily involved in the formations of ligand–protein hydrogen bonds, underwent minimal fluctuations for all the five compounds (Appendix A). Similarly, the average RMSF values of CTP synthase when complexed with the 5 compounds ranged from 0.92 to 1.32 Å (Figure 4C, Appendix A). Residues Ala59, Lys63, Glu64, Ile65, Val301, Tyr295, Tyr302, Gly349, Phe351, Arg453, and Glu501, which were crucial for the formations of protein-ligand hydrogen bonds and π–π stacking interactions, remained stable throughout the simulation (Appendix A). The average RMSF values of the 5 compounds complexed with TK ranged from 1.03 to 2.85 Å (Figure 5C, Appendix A). However, the average RMSF value of ZINC000031750813 (2.85 Å) was higher than that of the 4 other compounds selected against TK. For all five of the compounds, residues 205–215 of TK underwent fluctuations during the simulation. However, the residues that were crucial for the formations of ligand–receptor hydrogen bonds and π–π stacking interactions, including Pro10, Ser12, Cys13, Gly14, Lys15, Thr16, Phe42, Arg45, Tyr46, Glu86, Phe89, and Gly184, remained stable throughout the simulation (Appendix A). The average RMSF values of the 5 compounds complexed with DHFR ranged from 1.21 to 1.67 Å (Figure 6C, Appendix A), while the average RMSF values of MetAP2 complexed with the selected 5 compounds ranged from 0.68 to 0.79 Å (Appendix A). Additionally, residues 16–23, 175–184, and 248–258 underwent fluctuations during the simulation. However, the amino acid residues that mediated the formations of protein-ligand hydrogen bonds and π–π stacking interactions remained stable during the simulation (Figure 7C, Appendix A).

#### 2.7.4. Dynamic Cross-Correlation and PCA

The cross-correlation matrix of the resulting trajectories was analyzed for understanding the dynamical correlations of conformational motion of the protein-ligand complexes. The positive regions of the matrix are associated with the strongly correlated motions of residues moving in the same direction, while negative regions are linked with anti-correlated movements. The correlations of motion of the residues of AQP complexed with the five ligands were generated and displayed as a correlation matrix, depicted in Appendix A. Deeper colors indicate more positively or negatively correlated motions between the structural patterns. The pink blocks displayed in the figure indicate residues with highly correlated movements, while the green blocks indicate the least correlation. Residues 10–100, 125–150, and 175–200 of AQP complexed with CHEMBL133039, CHEMBL3140193, and CHEMBL3703838 exhibited concerted movements. These residues included amino acids that formed the binding pocket of AQP. The MD simulation trajectories of AQP complexed with the five compounds were subjected to a PCA, and the results are depicted in Appendix A. A dynamic cross-correlation matrix of CTP synthase complexed with the five compounds was similarly generated (Appendix A). Analysis of the resulting matrix revealed that residues 1–100, 150–250, and 275–400 of CTP synthase complexed with CHEMBL525202, CHEMBL1091856, and CHEMBL1162979 exhibited considerable correlated movements. The results further indicated that 72% of the amino acids in the 275–400 residue stretch were involved in the formation of the ligand-binding pocket. The dynamical cross-correlated maps of TK complexed with the five different compounds are graphically presented in Appendix A. Residues 1–50, 60–140, and 150–200 of TK exhibited a strong positive correlation when complexed with ZINC000031750813, CHEMBL3674540, and CHEMBL4078273, but the correlation decreased when bound to CHEMBL391279 and CHEMBL1683320. The cross-correlation matrix of DHFR complexed with five compounds are depicted in Appendix A. The cross-correlation matrix of MetAP2 complexed with the five compounds revealed a strong correlation for residues 1–150 (Appendix A). However, residues 150–225 and 240–300 of MetAP2 exhibited reduced correlation when complexed with CHEMBL3913373. Notably, these regions comprised amino acids that formed the ligand-binding pocket of MetAP2.

### 2.8. Determination of Binding Free Energies of Protein-Ligand Complexes

The binding free energy represents the sum total of all the interaction energies, including the van der Waals energy, polar solvation energy, electrostatic energy, and solvent-accessible surface area SASA energy. The binding free energies of all the complexes were estimated using the MM/GBSA approach (Table 4). The binding free energies of the 5 compounds complexed with AQP ranged from −2.8214 to −36.0654 kcal/mol, of which CHEMBL3703838 (−36.0654 ± 2.6122 kcal/mol) had the lowest free energy of binding. Further analysis revealed that van der Waals interactions played a major role in stabilizing the docked complexes (Appendix A). The intermolecular protein-ligand interactions after the MD simulation are depicted in Appendix A. The free energies of binding of the 5compounds complexed with CTP synthase ranged from −37.1662 to −56.5194 kcal/mol (Table 4), of which CHEMBL1091856 had the lowest binding free energy of −56.5194 ± 5.0206 kcal/mol. Further analysis revealed that van der Waals and electrostatic interactions played a major role in stabilizing the docked complexes (Appendix A). The intermolecular protein-ligand interactions of CTP synthase at the end of the production run are depicted in Appendix A. The free energies of binding of the 5 compounds selected against TK ranged from −9.1083 to −33.5752 kcal/mol, of which CHEMBL4078273 had the lowest free energy of binding of −33.5752 ± 4.6211 kcal/mol (Table 4). The interaction energies of all the five compounds selected against TK are depicted in Appendix A, and the results demonstrated that van der Waals and electrostatic interactions played a major role in stabilizing the docked complexes. The intermolecular protein-ligand interactions of TK at the end of the production run are depicted in a 2D plot (Appendix A). The free energies of binding of the 5 compounds complexed with DHFR ranged from −28.6175 to −35.9711 kcal/mol, of which CHEMBL340488 had the lowest free energy of binding of −35.9711 ± 3.1254 kcal/mol (Table 4). The van der Waals interactions between the compounds and DHFR played a major role in stabilizing the protein-ligand complexes (Appendix A). The intermolecular protein-ligand interactions of DHFR are depicted in a 2D plot (Appendix A).The free energies of binding of the 5 compounds selected against MetAP2 ranged from −28.6175 to −43.5796 kcal/mol (Table 4), of which CHEMBL3913373 had the lowest free energy of binding of −43.5796 ± 5.1314. The interaction energies of the five compounds complexed with MetAP2 are depicted in Appendix A, and the intermolecular protein-ligand interactions of MetAP2 at the end of the production run are depicted in Appendix A.

## 3. Discussion

The slow growth syndrome of shrimp caused by EHP infections is a major threat to sustainable shrimp farming in Asia at present. The disease has caused significant economic losses for shrimp farmers not only in Thailand but also in other parts of Southeast Asia since 2009. Shinn (2019) [15] calculated that EHP infections resulted in a loss of USD 387.9 to USD 555.8 million in 2019 owing to the reduction in harvest size from 18 g to 13 g and an increase in production costs of 23.2%. EHP infections have been recently spreading to new geographical locations in Korea [16], Venezuela [4], and Mexico, and there are no effective methods for limiting the negative effect of EHP infections on shrimp cultivation to date. In this study, we identified drug-like molecules against five potential druggable target proteins of EHP, namely, AQP, CTP synthase, DHFR, MetAP2, and TK, by employing structure-based VS and MD simulations. The selected molecules are biologically active compounds listed in the National Center for Advancing Translational Sciences (NCATS). Detailed information on these drug-like compounds was included in Appendix A.

CTP synthase catalyzes the synthesis of cytidine 5′-triphosphate (CTP) from uridine 5′-triphosphate (UTP) in the final step of the production of cytidine nucleotides from both the de novo and uridine salvage pathways [17]. CTP synthase is precisely regulated by intracellular concentrations of CTP and UTP. As a consequence, CTP synthase activity regulates the intracellular rates of RNA, DNA, and phospholipid synthesis [18]; therefore, it has been selected as a target for the development of drugs. CTP is not only a building block for nucleic acids but is also required for protein glycosylation [19], lipid biogenesis [20], and cellular communication [21]. Inhibition of the CTP synthase enzyme of *T. gondii* disrupts the lytic cycle [17]. CTP synthase is also a potential target for drug discovery against *T*. *brucei* [22], malaria [23], and giardiasis [24]. A structural analysis revealed that CTP synthase comprises an ammonia ligase (AL) domain and a glutamine amidotransferase (GAT) domain and also catalyzes the final step of de novo CTP biosynthesis [25]. The GAT domain catalyzes the hydrolysis of glutamine to yield NH_3_, which is subsequently utilized for the ATP-dependent conversion of UTP to CTP. A structural analysis of *Drosophila* CTP synthase (dmCTP synthase) revealed that the guanine base of GTP interacts with the Leu107 and Leu444 of another monomer. Further studies revealed that GTP forms three hydrogen bonds with Arg481. In addition, the π–π interaction between the guanine ring and Phe373 also contributes to the binding. The ribose ring of GTP interacts with Phe50 and Arg479 via hydrogen bonds. The triphosphate moiety of GTP also forms three hydrogen bonds with Lys306, Tyr307, and Arg376. The binding of glutamine via interactions with Phe37 and Cys399 further stabilizes the binding of GTP [25]. A comparison of the sequence and structure of EHP CTP synthase with dmCTP synthase resulted in the identification of similar important amino acid residues, including Tyr295, Phe351, Cys377, and Arg453, in the binding pocket of EHP CTP synthase. The results of the molecular docking revealed that all five of the compounds interacted with Phe351, Arg453, and Tyr295 via hydrogen bonds and π–π interactions in the binding pocket, which are equivalent to residues Phe373, Arg481, and Tyr307, respectively, of dmCTP synthase. Further analysis revealed that three compounds, namely, CHEMBL1091856, CHEMBL1162979, and CHEMBL525202, had the lowest free energies of binding with EHP CTP synthase. The compound 6-diazo-5-oxo-L-norleucine (DON) is a glutamine analog that inhibits the functions of CTP synthase. DON interacts with Phe373 and Cys399 and covalently binds to the active site of glutaminase [25]. Potent inhibitors of the CTP synthase enzymes of *Plasmodium* sp., *Trypanosoma* sp., and *Toxoplasma* sp., includingα-amino-3-chloro-4,5-dihydro-5-isoxazoleacetic acid, glutamate γ-semialdehyde, azaserine, and 2,4-diaminopentanedioic acid, have been identified and reported in previous studies [22,26,27].

The DHFR catalyzes the NADPH-dependent reduction of dihydrofolate (DHF) to tetrahydrofolate (THF). Tetrahydrofolates serve as co-factors for the synthesis of purines, pyrimidine (thymidylate), and for the re-methylation of homocysteine to methionine. Reduction in DHFR enzymatic activity diminishes the THF pool inside the cell, which slows DNA synthesis and cell proliferation, eventually leading to cell death [28,29]. DHFR is a potential druggable target owing to its metabolic importance. DHFR inhibitors are commonly used for the treatment of malaria [11] and other protozoal infections, including *T*. *cruzi* [30] and *Cryptosporidium hominis* [30]. Several classes of compounds with potential antifolate activity are used in the treatment of cancer and rheumatoid arthritis (methotrexate, MTX), bacterial DHFR enzyme (trimethoprim, TMP), and the DHFR of *P*. *falciparum* (pyrimethamine, PYR). Although pyrimidine biosynthesis occurs ubiquitously in EHP and shrimp, the DHFR enzyme is divergent enough to allow the design of inhibitors specific to EHP DHFR. In this study, we identified five compounds with high binding affinity to the folate binding pocket of EHP DHFR. Of these compounds, ZINC000016682972 had the lowest free energy of binding and exhibited high stability with EHP DHFR. Further analysis revealed that MTX and TMP bind to the same region of DHFR, which was selected as the druggable site for docking-based VS against EHP DHFR in this study. DHFR inhibitors, including riluzole, 6-(trifluoromethoxy)-1,3-benzothiazol-2-ylamine, 7-((2-thiazol-2-yl)benzimidazol-1-yl)-2,4 diamino quinazoline, and proguanil, which specifically bind to the folate-binding pocket of DHFR, have been identified for suppressing infections caused by *L*. *major*, *T*. *brucei*, *Staphylococcus aureus*, and the malarial parasite, respectively [31,32,33].

MetAP2 catalyzes the removal of N-terminal methionine residues from nascent proteins [34]. Inhibition of MetAP activity could therefore affect protein biological activity and proper cellular localization and turnover, and result in interference with cell signal transduction and cell-cycle progression [35]. The microsporidia contain only MetAP2 in their genome which makes microsporidia MetAP2 an essential target for designing therapeutic agents for microsporidiosis [36,37]. The MetAP2 protein has been identified as a potential target for the treatment of infections caused by *E*. *bieneusi*, *E*. *cuniculi*, *Entamoeba histolytica,* and *Vittaforma corneae* [37,38,39]. A gene-encoding MetAP2 is also present in EHP (EhpMetAP2), and EhpMetAP2 shares 51% identity with the MetAP2 protein of *E*. *cuniculi* (EcMetAP2). A structural analysis of EcMETAP2 revealed that it has a metal-binding domain and a methionine-binding domain. Residues Asp130, Asp141, His210, Glu243, and Glu339, which are responsible for coordinating Fe^2+^ ions, were found to be completely conserved in all the MetAP protein sequences, including EcMetAP2.

The residues comprising the methionine-binding pocket include Phe97, Pro98, His109, Ile217, and His218 inthe peripheral loops and His261, Val263, Pro292, and Tyr324 in the subdomain. Only two residues, His109 and His218, are completely conserved across the entire MetAP protein family [38]. The methionine-binding domain identified in EhpMetAP2 comprised similar conserved amino acid residues, including Phe80, Pro81, His92, Ile201, His202, His244, Pro275, and Tyr304, which formed a druggable pocket that was targeted with docking-based VS in this study. The results of molecular docking revealed that all five of the compounds formed hydrogen bonds with His92 and His202, which are equivalent to His109 and His218 of EcMetAP2. Further analysis revealed that CHEMBL3913373, CHEMBL1962731, and ZINC000199197855 formed π–π interactions with His92, His202, and Tyr304. The free energies of binding of ZINC000016682972, CHEMBL3142997, and CHEMBL1962731 with EhpMetAP2 were the lowest among all the selected compounds.

Fumagillin/TNP-470 is a well-known inhibitor of MetAP2 that binds to the methionine-binding pocket of EcMETAP2 via hydrophobic interactions with Phe97, Pro98, Ilu217, His261, Val263, Pro292, and Tyr324, which are equivalent to residues Phe80, Pro81, Ile201, His244, Met246, and Tyr304 of EhpMetAP2 [38]. However, the fumagillol core of fumagillin/TNP-470 forms hydrophobic interactions with the conserved residues His231, Leu328, Val374, and Leu447 and forms a single hydrogen bond with Asp376 of human MetAP2b (HsMetAP2b), which are equivalent to residues His92, Leu191, Glu253, Leu307, and Glu238 of EhpMetAP2. The unsaturated decanoic acid side chain of fumagillin is stabilized via interactions with Asn327, Asn329, and His375 of HsMetAP2b, which are equivalent to residues Asn190, Asn192, and Lys238 of EhpMetAP2 [38]. Apart from fumagillin, some other potential inhibitors of HsMetAP2b, including spiroepoxytriazole, bestatin, amastatin, and 3-anilino-5-benzylthio-1,2,4-triazole, have been reported in previous studies [40,41].

The TK key enzyme catalyzes the transfer of the γ-phosphate of ATP to 2′-deoxythymidine (dThd), forming thymidine monophosphate (dTMP) via the salvage pathway in microsporidia [42]. Inhibition of TK causes severe dTTP depletion that leads to the massive incorporation of uracil into DNA and contributes to the phenomenon called “thymineless death” [43]. TK is a potential druggable target of *L*. *major* [44] and *T*. *brucei* [45]. The gene encoding the TK enzyme is also present in EHP. Human TK1 (hTK1) is the most studied type II enzyme that consists of a Zn-binding domain, an ATP-binding domain, a 20-dThd-thymidine-binding domain, and a lasso domain. The primary sequence of EHP TK (EhpTK) significantly differs from the human TK (hTK) enzyme. However, a sequence-based comparison of the active sites of EhpTK and HuTK1 revealed that the majority of amino acids surrounding the deoxyribonucleoside moiety are identical in both enzymes. Tyr163 of HuTK1 is part of a hydrophobic pocket that surrounds the 5-methyl group of thymine. The 5-methyl group is also surrounded by residues Leu124, Tyr181, and Met28, which are equivalent to Thr153, Leu114, Tyr189, and Val11 of EhpTK. The ribose moiety of dThd is stabilized via interactions with Asp88 and Glu98, which are equivalent to residues Asp43 and Glu86 of EhpTK [46]. Glu98, which is essential for catalysis, forms a hydrogen bond with the 5′-oxygen of the ribose ring, and is well-placed to act as the catalytic base for abstracting a proton from the oxygen. This enables the oxygen atom to perform a nucleophilic attack on the γ-phosphate of the phosphate donor [44,47]. The results of molecular docking revealed that all the compounds bound stably to the binding pocket via hydrogen bonds with Glu86 and Arg45. However, CHEMBL4078273, CHEMBL1683320, and CHEMBL3674540 had the lowest free energies of binding and formed highly stable complexes with EhpTK. Several thymidine analogs, including aurantiamide acetate, zidovudine, stavudine, azidothymidine, and 3-trifluoromethyl-4-chloro-phenyl-urea-α-thymidine, have been previously identified for the treatment of infections caused by *P*. *falciparum*, *Giardia intestinalis*, and *T*. *cruzi* [48,49,50].

AQPs are transmembrane channels that transport water and/or small solutes, such as glycerol, nitrates, and urea, across cellular membranes. A total of 13 AQP isoforms (AQP0 to AQP12) have been identified in humans [51]. Of these, AQP3, AQP7, AQP9, and AQP10 are aquaglyceroporins that facilitate the transport of glycerol and other small neutral solutes such as urea, ammonia, and carbon dioxide [52]. The solute selectivity of AQPs is determined by two-channel sections, namely, the conserved asparagine–proline–alanine (NPA) region and the aromatic/arginine (ar/R) constriction [53]. A comparison of the sequences and structures of the AQP protein of EHP and human AQP1 revealed the presence of a conserved NPA region in the AQP protein of EHP. However, the alignment revealed that an arginine (Arg195) was substituted for an isoleucine (Ile204) in the AQP protein of EHP, which clearly indicated that it belongs to the aquaglyceroporin subset. Arg195 is conserved in all members of the AQP superfamily that are selective for water transport [54]. It was postulated that aquaporin transports water inside spores, and as a result, the osmotic pressure quickly increases inside the spore which triggers the shooting out of its polar tube and transferring its sporoplasm into a host cell [55,56]. The inhibition of AQPs with HgCl_2_ effectively inhibits the germination of *Anncaliia algerae* spores [57]. The AQP protein of *A*. *algerae* was, therefore, identified as a potential druggable target in an earlier study. [57]. In this study, we identified the conserved NPA region and the residues in the selective filter region of the AQP protein of EHP by comparing the sequence with those of human AQP1, AQP3, and AQP4 proteins. The binding pocket formed by these residues was subsequently targeted for screening drug molecules [58,59]. A similar region was selected in a study by Yadav et al. in 2020 [59] for screening drug molecules against the AQP3 protein of humans. The results demonstrated that all the docked poses formed several direct hydrogen bonds with important residues—including Asn60 and Arg218—while the backbone atoms of Gly145, Ala148, Gly207, Gly211, and Phe208 were involved in the formation of protein-ligand hydrogen bonds. Additionally, the majority of compounds formed π–π stacking interactions with the aromatic rings of Tyr150 and Phe208. Another study demonstrated that acetazolamide, a carbonic anhydrase inhibitor, reduces the water permeability of the AQP1 protein from the oocytes of *Xenopus laevis* by binding to a region similar to that identified in the present study [60].

## 4. Materials and Methods

### 4.1. Identification of Potential Druggable Target Proteins

The potential druggable protein targets of EHP were identified using two different approaches. The complete protein information of EHP was retrieved from the UniProt Knowledgebase and subjected to a functional enrichment analysis for identifying the candidate proteins that are involved in major metabolic pathways, using the online tools in DAVID for a KEGG enrichment analysis. The protein targets were finally selected using an approach similar to that used in earlier studies [61,62], which involved a thorough review of the literature published on microsporidians [17,22,26,37,38,45].The amino acid sequences of the five druggable protein targets of EHP, including AQP, CTP synthase, MetAP, DHFR, and TK, were retrieved from the UniProt Knowledgebase (https://www.uniprot.org/ accessed on 5 January 2022). The three-dimensional structures of all the five proteins were generated using AlphaFold 2 [63]. The stereochemical qualities of the protein models were further validated with the ProSA and PROCHECK modules of the PDBsum server [64].

### 4.2. Identification of Druggable Pockets

The druggable pockets in the five protein targets were identified using two separate approaches. The amino acid sequences of the five proteins were first used for identifying all the sequence homologs in the PDB with the NCBI BLAST server [65], using default parameters. The sequences of the closely related protein homologs were retrieved, and a multiple sequence alignment was performed for identifying the consensus and conserved residues in the ligand-binding sites of the proteins across the different members using CLC Workbench v8.5. software (Qiagen). The ligand-binding pockets were subsequently predicted using the CASTp server [66]. The appropriate target sites were finally selected based on previous knowledge of the conserved residues lining the pockets.

### 4.3. Structure-Based VS

The three-dimensional structures of the five proteins of EHP were used as targets for screening drugs from the ZINC15 [67] and ChEMBL [68] databases with docking-based VS using the EasyVS web-based VS tool (http://biosig.unimelb.edu.au/easyvs/ accessed on 25 January 2022) [69]. A sequential screening strategy was employed for screening the drug molecules. The drug molecules were initially filtered via Lipinski’s rule of five, and the selected molecules were docked using the web-based EasyVS tool. A chemical space was subsequently prepared around the druggable site that was selected as the potential target site (Table 1).

### 4.4. Prediction of ADMET Properties

All the screened molecules were subjected to an ADMET analysis for predicting the pharmacokinetics and toxicity properties using the SwissADME server (https://admetmesh.scbdd.com/ accessed on 10 August 2022) [70]. ADMET studies provide insights into various pharmacokinetic properties, including absorption, distribution, metabolism, excretion, and toxicity.

### 4.5. Screening Based on EC

The compounds were initially screened based on the following criteria: affinity scores ≥ 7.5 and numbers of interactions ≥ 9. These compounds were re-screened based on their EC [71]. The results were further enriched via re-docking the selected compounds with Flare v5.0.0 (Cresset Inc., Cambridge, UK), and compounds with high EC values were selected for further studies. The selected compounds were additionally screened using the PatchDock server [72].

### 4.6. MD Simulations of Protein-Ligand Complexes

The stabilities of the protein-ligand complexes were verified with 100 ns MD simulations using the Amber ff19SB [73] force field and the general AMBER force field (GAFF) [74]. A dodecahedral box of 12 Å was constructed around the protein-ligand complexes, and the box was solvated with TIP3P water. The excess charges were neutralized via the addition of either Na^+^ or Cl^−^ ions at a molar concentration of 0.15 M. The systems were subjected to energy minimization for relaxing the water molecules and intramolecular steric clashes at a temperature of 300 K under 1 bar pressure. The systems were subsequently equilibrated for 20,000 ps while imposing positional restraints of 700 kJ/mol. All the simulations were performed under the NPT ensemble by maintaining the temperature at 300 K using the Langevin thermostat [75], with a collision frequency of γ = 1/ps. Pressure control was achieved by coupling the system to a Monte Carlo barostat [76] at a reference pressure of 1 atm and a relaxation time of 2 ps. The simulations were performed using the GPU-accelerated version of the OpenMM 7.6 engine [77] and the ‘Making it rain’ [78] cloud-based molecular simulations notebook environment. The trajectories generated during the MD simulations of the protein-ligand complexes were analyzed for calculating the values of RMSD, RMSF, and hydrogen bonds using scripts included in AMBER. The trajectories were subjected to a PCA, and the cross-correlation maps of the entire trajectories were analyzed.

### 4.7. Determination of Free Energies of Protein-Ligand Complexes

The binding free energies of the docked complexes were calculated using the mechanics/generalized Born surface area (MM/GBSA) approach [79]. The binding free energies (ΔG_bind_) were calculated using the following equations [80,81]:ΔG_bind_ = ΔG_complex_ − (ΔG_receptor_ + ΔG_ligand_)(1)
where ΔG_complex_, ΔG_receptor_, and ΔG_ligand_ represent the free energy of the complex, receptor, and ligand, respectively.
ΔG = ΔE_gas_ + ΔG_sol_ − TΔS_gas_(2)
ΔE_gas_ = ΔE_int_+ ΔE_ELE_ + ΔE_VDW_(3)
ΔGsol = ΔGGB + ΔGSurf(4)
where ΔG represents the free energy. The energy in the gas phase (ΔE_gas_) comprises the internal energy (ΔE_int_), electrostatic interactions (ΔE_ele_), and van der Waals interactions (ΔE_vdw_) energy terms. The solvation free energy (ΔG_sol_) comprises the polar energy (ΔG_GB_) and non-polar energy (ΔG_Surf_) terms. TΔS_gas_ represents the contribution of conformational entropy.

## 5. Conclusions

EHP is an intracellular parasite that is responsible for the slow growth syndrome of the Penaeid shrimps *L. vannamei* and *P. monodon*. Shrimp production in Asia has declined by 10–20% owing to EHP infections, which has resulted in significant economic losses. In addition, the infection is rapidly spreading to new geographical locations. The shrimp farming industry will suffer substantial economic losses if the scenario of EHP infections remains unaltered, which will have serious effects on the global socio-economic structure. In this study, a total of fifteen compounds (CHEMBL3703838, CHEMBL2132563, CHEMBL133039, CHEMBL1091856, CHEMBL1162979, CHEMBL525202, CHEMBL4078273, CHEMBL1683320, CHEMBL3674540, ZINC000016682972, CHEMBL3142997, CHEMBL340488, CHEMBL1966988, ZINC000828645375, and CHEMBL1962731) were identified against five potential druggable protein targets of EHP. The compounds had high binding affinities and low free binding energies, as indicated by the results of extensive the insilico analyses. The compounds formed stable complexes with the respective protein targets and were predicted to have insilico inhibitory potentials. The results of the computational analyses obtained in this study will be experimentally validated by in vitro and in vivo studies in future.

## Figures and Tables

**Figure 1 ijms-24-01412-f001:**
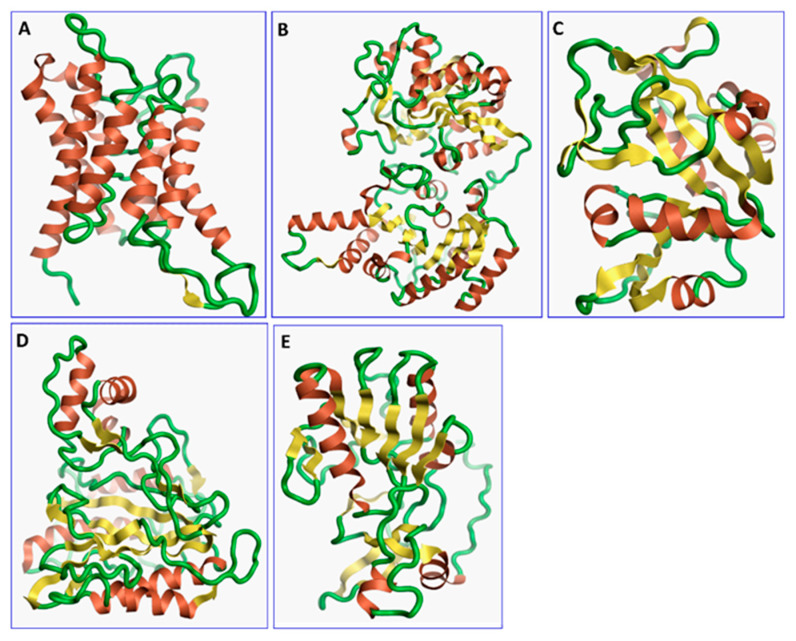
The three-dimensional structures of the five different protein targets—(**A**) AQP protein, (**B**) CTP synthase, (**C**) DHFR, (**D**) MetAP2, and (**E**) TK—of EHP generated using AlphaFold 2. The α-helices are depicted as red coils, while the β-strands are represented as yellow arrows.

**Figure 2 ijms-24-01412-f002:**
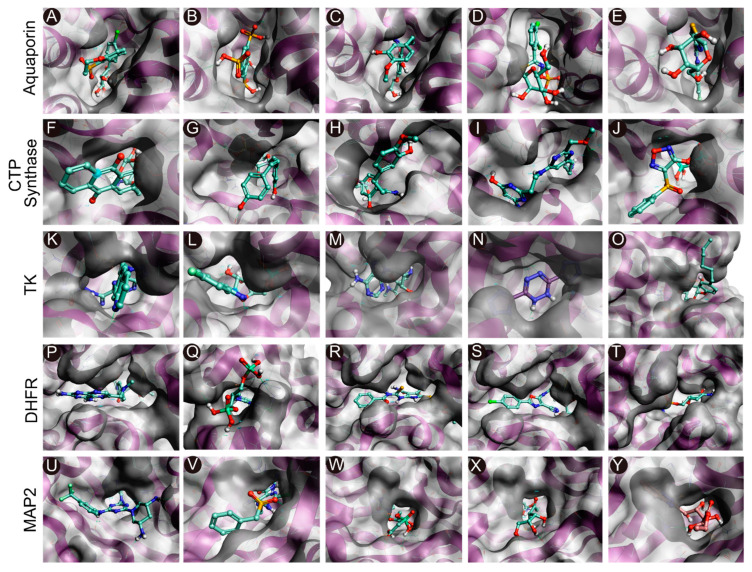
Molecular docking of the compounds to the predicted binding pockets of the five druggable protein targets of EHP. (**A**–**E**) Best binding poses of the five selected compounds atthe binding site of AQP ((**A**): CHEMBL3703838; (**B**): ZINC000002243083; (**C**): CHEMBL133039; (**D**): CHEMBL3140193; and (**E**): CHEMBL2132563). (**F**–**J**) Best binding poses of the five selected compounds atthe binding site of CTP synthase ((**F**): CHEMBL48494; **G**: CHEMBL1162979; (**H**): CHEMBL133039; (**I**): CHEMBL525202; and (**J**): CHEMBL1091856). (**K**–**O**) Best binding poses of the five selected compounds atthe binding site of TK ((**K**): CHEMBL3674540; (**L**): CHEMBL1683320; (**M**): CHEMBL391279; (**N**): ZINC000031750813; and (**O**): CHEMBL4078273). (**P**–**T**) Best binding poses of the five selected compounds atthe binding site of DHFR ((**P**): CHEMBL1966988; (**Q**): CHEMBL340488; (**R**): ZINC000016682862; (**S**): ZINC000828645375; and (**T**): CHEMBL3901573). (**U**–**Y**) Best binding poses of the five selected compounds at the binding site of MetAP2 ((**U**): CHEMBL3913373; (**V**): CHEMBL1962731; (**W**): CHEMBL3142997; (**X**): ZINC000199197855; and (**Y**): ZINC000016682972).

**Figure 3 ijms-24-01412-f003:**
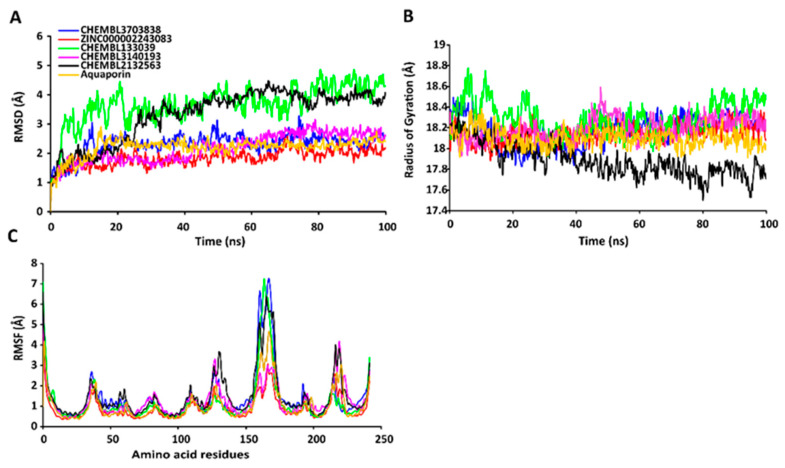
Stabilities of the protein-ligand complexes during 100 ns MD simulations. (**A**) RMSD values of the protein-ligand complexes fitted to the Cα backbone of AQP. The *x*-axis depicts the duration of simulation, while the *y*-axis represents the deviation. (**B**) Analysis of the Rg values of the protein-ligand complexes. The *x*-axis depicts the duration of simulation, while the y-axis represents the deviations in Rg. (**C**) RMSF values of the Cα backbone of AQP. The *x*-axis depicts the total number of residues, while the y-axis represents the RMSF in Å. The different ligands complexed with AQP are represented by lines of different colors.

**Figure 4 ijms-24-01412-f004:**
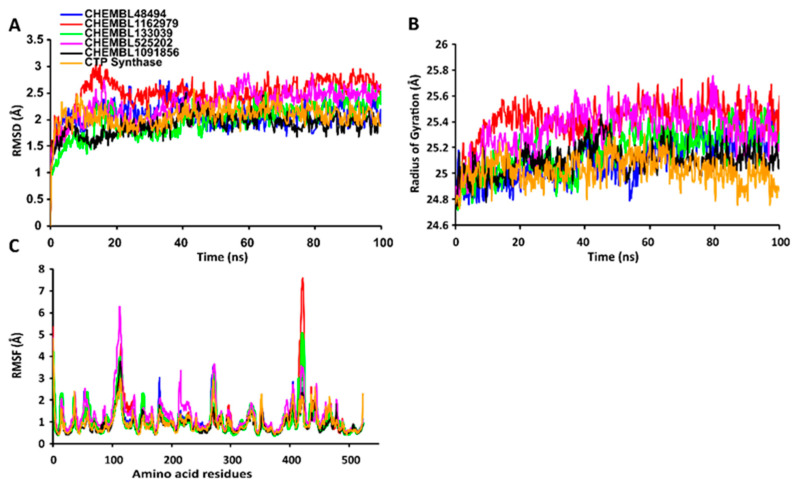
Stabilities of the protein-ligand complexes during 100 ns MD simulations. (**A**) RMSD values of the protein-ligand complexes fitted to the Cα backbone of CTP synthase. The *x*-axis depicts the duration of simulation, while the *y*-axis represents the RMSD. (**B**) Analysis of the Rg values of the protein-ligand complexes. The *x*-axis depicts the duration of simulation, while the *y*-axis represents the deviations in Rg. (**C**) The RMSF values of the Cα backbone of CTP synthase. The *x*-axis depicts the total number of residues, while the *y*-axis represents the RMSF in Å. The different ligands complexed with CTP synthase are represented by lines of different colors.

**Figure 5 ijms-24-01412-f005:**
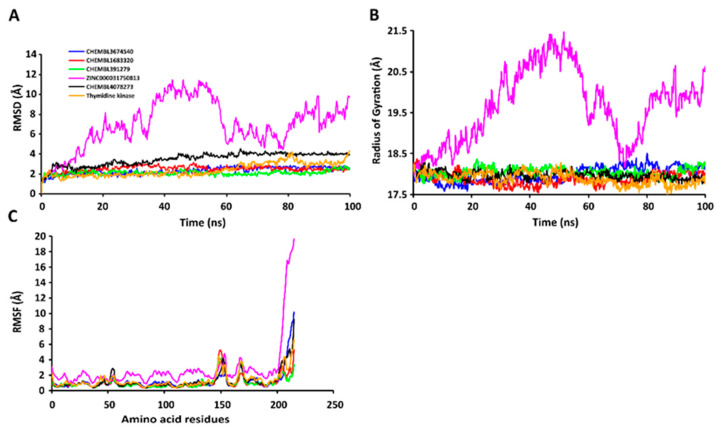
Stabilities of the protein-ligand complexes during 100 ns MD simulations. (**A**) RMSD values of the protein-ligand complexesfitted to the Cα backbone of TK. The *x*-axis depicts the duration of simulation, while the *y*-axis represents the RMSD. (**B**) Analysis of the Rg values of the protein-ligand complexes. The *x*-axis depicts the duration of simulation, while the *y*-axis represents the deviations in Rg. (**C**) RMSF values of the Cα backbone of TK. The *x*-axis depicts the total number of residues, while the *y*-axis represents the values of RMSF in Å. The different ligands complexed with TK are represented by lines of different colors.

**Figure 6 ijms-24-01412-f006:**
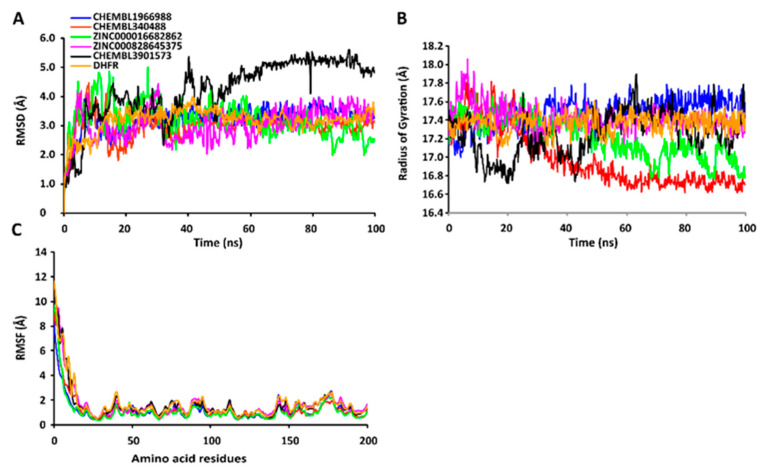
Stabilities of the protein-ligand complexes during 100 ns MD simulations. (**A**) RMSD values of the protein-ligand complexes fitted to the Cα backbone of DHFR. The *x*-axis depicts the duration of simulation, while the *y*-axis represents the RMSD. (**B**) Analysis of the Rg values of the protein-ligand complexes. The *x*-axis depicts the duration of simulation, while the *y*-axis represents the deviations in Rg. (**C**) RMSF values of the Cα backbone of DHFR. The *x*-axis represents the total number of residues, while the *y*-axis depicts the RMSF in Å. The different ligands complexed with DHFR are indicated by lines of different colors.

**Figure 7 ijms-24-01412-f007:**
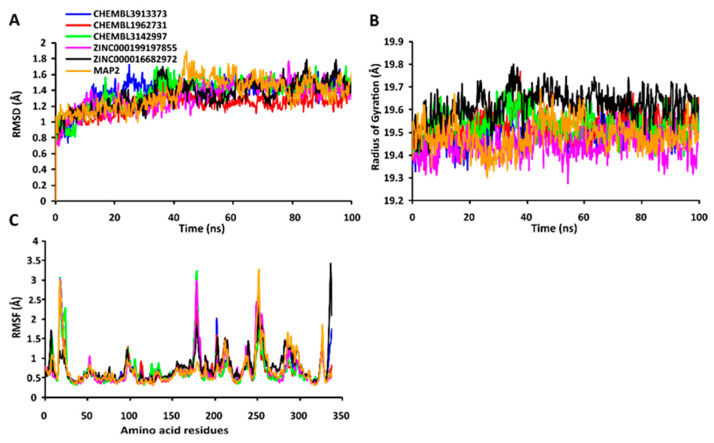
Stabilities of the protein-ligand complexes during 100 ns MD simulations. (**A**) RMSD values of the protein-ligand complexes fitted to the Cα backbone of MetAP. The *x*-axis represents the duration of simulation, while the *y*-axis depicts the RMSD. (**B**) Analysis of the Rgvalues of the protein-ligand complexes. The *x*-axis represents the duration of simulation, while the *y*-axis depicts the deviations in Rg. (**C**) RMSF values of the Cα backbone of MetAP. The *x*-axis depicts the total number of residues, while the *y*-axis represents the RMSF in Å. The different ligands complexed with MetAP are indicated by lines of different colors.

**Table 1 ijms-24-01412-t001:** Identification of binding pockets in the five druggable protein targets of EHP and generation of grid box for molecular docking studies.

Protein Name	Coordinate of Docking Box	Structure	Amino Acid Residues
Aquaporin	X: 4.7541; Y: 0.6196; Z: 0.8248	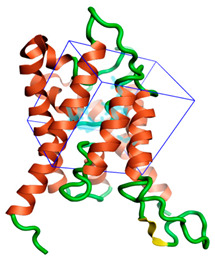	Phe22, Gly23, Val45, Val49, Glu138, Leu145, Gly198, Ala199, Phe200, Asn201, Pro202, Gly203, Ile204
CTP synthase	X: −11.6484; Y: −1.5365; Z: 7.2675	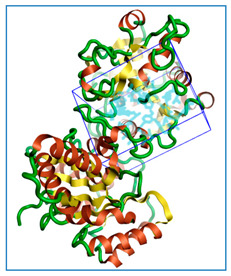	Ala59, Glu64, Ile65, Ile292, Thr293, Arg294, Tyr295, Val301, Tyr302, Leu305, Cys347, Pro348, Gly349, Gly350, Phe351, Gly352, Thr354, Lys359, Ile376, Cys377, Leu378, Arg453, His499, Glu502, Leu503
Thymidine kinase	X: 6.300; Y: −0.1442; Z: −3.7617	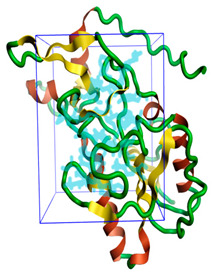	Val11, Ser12, Cys13, Gly14, Lys15, Thr16, Ilc17, Lys39, Asp43, Arg45, Tyr46, Ser50, Ile51, Lys52, Ser53, Ala54, Asp85, Glu86, Gln88, Phe89, Gly113, Leu114, Lys116, Asp117, Phe118, Phe123, Ser161, Lys180, Cys183, Gly184, Gly185, Ile186, Tyr189
Dihydrofolate reductase	X: 3.5346; Y: 3.8013;Z: 4.8650	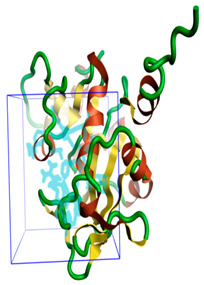	Leu27, Val28, Ala29, Ile37, Ser38, Gly40, Glu41, Lys42, Met43, Trp45, Arg47, Leu48, Ser4, Asp51, Phe52, Ala53, Met55, Lys56, Met59, Gly70, Arg71, Lys72, Thr73, Glu75, Val76, Ala77, Lys78, Tyr79, Thr80, Asn81, Tyr82, Leu87, Ser88, Arg89, Lys102, ser103, Phe104, Ala119, Gly120, Thr138, Arg150
Methionine Aminopeptidase 2		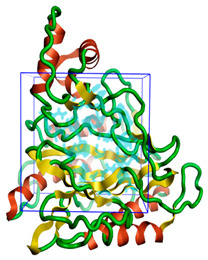	Asn75, Asn76, Gly77, Ile78, Gly79, Phe80, Pro81, Gly83, Ser85, Ala90, Ala91, His92, Lys111, Asp113, Asp124, Leu191, His194, Ile201, His202, Glu226, Phe228, His224, Phe245, Met246, Pro275, Pro303, Try304, Pro305, Pro306, Leu307, Gln317, Glu319

**Table 2 ijms-24-01412-t002:** Total amino acid residues present in the binding pockets of different proteins of EHP and *L. vannamei*.

Protein Name	*E. hepatopenaei*	*L. vannamei*
Aquaporin	Phe22, Gly23, Val45, Val49, Glu138, Leu145, Gly198, Ala199, Phe200, Asn201, Pro202, Gly203, Ile204	**A0A193KUU7**: Leu36, Val37, Ile59, Phe63, Glu149, Pro156, Pro200, Ala201, Arg 202
		**A0A3R7N2N5**: Glu60, Leu67, Phe131, Cys232, Phe233, Pro234, Pro265, Asn226, Pro267
		**A0A3R7PEU6**: Met26, Gly30, Tyr112,Trp119
		**A0A3R7Q089**: Phe25, Gly26, Trp49, Gln156, Leu163, Ala284, Ser285, Leu286, Gly287
CTP synthase	Ala59, Glu64, Ile65, Ile292, Thr293, Arg294, Tyr295, Val301, Tyr302, Leu305, Cys347, Pro348, Gly349, Gly350, Phe351, Gly352, Thr354, Lys359, Ile376, Cys377, Leu378, Arg453, His499, Glu502, Leu503	Phe64, Gly69, Val70, Val317, Gly318, Lys319, Tyr320, Ser326, Tyr327, Val330, Val382, Pro383, Gly384, Ile385, Gly386, Gly387, Arg389, Lys394, Val411, Cys412, Leu413, Gly492, Val530, Tyr 532,Val534
Dihydrofolate reductase	Leu27, Val28, Ala29, Ile37, Ser38, Gly40, Glu41, Lys42, Met43, Trp45, Arg47, Leu48, Ser4, Asp51, Phe52, Ala53, Met55, Lys56, Met59, Gly70, Arg71, Lys72, Thr73, Glu75, Val76, Ala77, Lys78, Tyr79, Thr80, Asn81, Tyr82, Leu87, Ser88, Arg89, Lys102, Ser103, Phe104, Ala119, Gly120, Thr138, Arg150	Val2, Tyr3, Ile15, Ala16, Asn19, Asn20, Glu21, Leu22, Trp24, His26, Glu27, Gly33, Asp35, Phe36, Gly37, Ser39, Ala40, Gln43, Gly51, Arg52, Lys53, Thr54, Asp56, Val58, Ala59, Gly60, Phe61, Asp62, Pro66, Tyr67, Phe72, Val73, Leu74, Lys88, Val89, Phe90, Ala101, Gly107, Try108, Asn109, Glu110, Leu111, Tyr112, Asp114, Pro145
Methionine aminopeptidase	Asn75, Asn76, Gly77, Ile78, Gly79, Phe80, Pro81, Gly83, Ser85, Ala90, Ala91, His92, Lys111, Asp113, Asp124, Leu191, His194, Ile201, His202, Glu226, Phe228, His224, Phe245, Met246, Pro275, Pro303, Try304, Pro305, Pro306, Leu307, Gln317, Glu319	**A0A423SS39**: Lys201, Ala202, Gly203, Leu204, Ala205, Phe206, Pro207, Gly209, Ser211, Ala216, Ala217, H218, Lys238, Asp240, Asp251, Leu315, His318, Ile325, His326, Glu351, Tyr370, Met371, Ala401, Pro430, Tyr431, Pro432, Pro433, Leu434, Gln317, Glu446
		**A0A423SV11**: His3, Ile10, His11, Glu36, Tyr55, Met56, Ala73, P102, Tyr103, Pro104, Pro105, Leu106, Gln116, Glu118
		**A0A423SSE3**: Asn176, Try177, His178, Gly179, Phe180, Phe181, Ser183, Ser187, Ile192, Cys193, His194, Asn209, Asp211, Phe219, His220, Trp331, Pro332, Gln347, Glu349
		**A0A423TUC3**: Leu79, Tyr80, His81, Gly82, Phe83, Pro84, Ser86, Ser90, Ile95, Cys96, His97, Asn112, Asp114, Phe122, His123, Gln196, Leu198
		**A0A423SU95**: Leu148, Asn149, Tyr150, His151, Gly152, Phe153, Pro154, Ser156, Ser160, Ala165, Cys166, His167, Asn182, Asp184, Tyr192, His193, Gly203, Glu222, Ala223, Tyr271, Gly284, Thr286
Thymidine kinases	Val11, Ser12, Cys13, Gly14, Lys15, Thr16, Ilc17, Lys39, Asp43, Arg45, Tyr46, Ser50, Ile51, Lys52, Ser53, Ala54, Asp85, Glu86, Gln88, Phe89, Gly113, Leu114, Lys116, Asp117, Phe118, Phe123, Ser161, Lys180, Cys183, Gly184, Gly185, Ile186, Tyr189	Gly4, Lys5, Thr6, Thr7, Asp31, Arg33, Tyr34, Gly38, Ile39, Ala40, Thr41, His42, Asp78, Thr79, Glu81, Pro107, Arg146, Phe159, Glu161, Vol162, Gly164, Ser166, Tyr169

Uniprot ID of proteins of *L. vannamei* written in bold.

**Table 3 ijms-24-01412-t003:** Binding affinities, EC scores, and interactions between the five selected compounds and the corresponding protein targets.

	Easy Vs			Flare						
Molecular ID	Affinity Score	Total Number of Bonds	PatchDock Score	Electrostatic Complementarity Score	Electrostatic Complementarity (Pearson R)	Electrostatic Complementarity (Spearman’s Rho)	Number of Hydrogen Bonds	Amino Acids Involved in Hydrogen Bonds (Bond Distance Å)	Number of π–π Bonds	Amino Acids in π–π Bond(s) (Bond Distance Å)
Aquaporin										
CHEMBL3703838	−8	9		0.286	0.337	0.335	5	Gly197 (1.8), Gly198 (2.9), Ala 199 (1.8), Lys34 (2.7) (2.7)	1	Thr121 (2.6)
ZINC000002243083	−7.6	9		0.302	0.451	0.325	6	Gly198 (2.4) (1.7), Thr33 (1.8), Ala123 (2.5) (2.6), Ser196 (2.2)		
CHEMBL133039	−7.6	9		0.337	0.448	0.549	5	Gly198 (2.1 Å), Gly197 (2.1 Å), Leu119 (2.5 Å), Thr33 (1.8 Å), Ile120 (1.7 Å)	1	Lys34 (3.7)
CHEMBL3140193	−7.5	9		0.295	0.361	0.341	6	Gly189 (1.9), Gly46 (3.0), Ser193 (1.7) (2.7), Lys34 (2.5), Thr33 (2.2)	1	Thr33 (3.9)
CHEMBL2132563	−7.5	9		0.289	0.303	0.311	5	Gly197 (2.6), Ser193 (3.0), Lys34 (2.6), Thr33 (1.9) (2.4)		
CTPsynthase										
CHEMBL48494	−9.6	11	5011	0.331	0.249	0.284	7	Arg453 (2.2), Ile65 (2.0), Tyr295 (2.2), Gly349 (2.6), Glu501 (2.0) (1.7), Gly349 (2.3)	3	Phe351 (3.1) (3.1), Arg453 (3.1)
CHEMBL1162979	−8.6	16	5250	0.301	0.467	0.459	13	Gly55 (1.7), Ala59 (2.6), Ile65 (2.5) (2.9), Glu64 (2.3), Glu501 (1.8), Gly349 (2.3), Arg453 (2.8) (3.0) (2.3) (2.1), His499 (2.8), Gly350 (2.5)	2	Arg453 (4.1), Phe351 (3.9)
CHEMBL133039	−8.3	12	5496	0.296	0.409	0.421	8	Ile65 (2.1), Phe351 (1.9) (2.7), Tyr295 (2.4), Val301 (3.4), Glu64 (2.1), Gly349 (2.0) (2.4)	3	Phe351 (3.0), Arg453 (4.3), Tyr302 (3.3)
CHEMBL525202	−8.1	11	5494	0.291	0.367	0.429	10	Phe351 (2.6), Vol301 (2.9), Ile65 (2.6) (1.9) (2.6), Arg453 (2.3) (2.6) (2.2) (2.4), Tyr295 (1.9)		
CHEMBL1091856	−7.6	17	5128	0.297	0.408	0.364	15	Ala59 (2.3), Tyr295 (2.2), Arg453 (2.9) (2.2) (2.1) (2.5), Phe351 (2.6) (2.4), Ile65 (1.8) (1.8), Glu64 (2.4) (1.9), Glu501 (1.8), Gly349 (2.8), His499 (2.9)	1	Phe351 (2.9)
Thymidine kinase										
CHEMBL3674540	−9.5	11	5118	0.286	0.389	0.433	7	Glu86 (2.5) (2.7) (2.1), Lys15 (2.5), Gly14 (2.3), Ala54 (2.0), Ile186 (2.2),		
CHEMBL1683320	−8.6	15	4888	0.267	0.352	0.381	13	Ser12 (1.9) (2.7), Gly14 (3.0), Arg45 (2.4) (2.0), Glu86 (2.8) (1.9) (1.9), Thr16 (2.8) (2.1) (2.2), Ile17 (2.2), Ala54 (2.3)		
CHEMBL391279	−8.3	16	4702	0.329	0.461	0.521	12	Ile186 (2.7), Tyr46 (3.0) (2.4), Arg45 (3.0), Glu86 (2.1) (2.9), Thr16 (1.8), Glu86 (2.1) (2.9), Lys15 (2.1) (2.4), Pro10 (2.9), Cys13 (2.8)	1	Arg45 (4.9)
ZINC000031750813	−8.2	14	3210	0.308	0.531	0.449	10	Gly14 (2.6), Ile17 (3.0), Ala54 (2.5), Thr16 (2.4) (2.3), Ser12 (2.1) (2.3), Gly86 (2.0) (2.3) (2.0)	1	Arg45 (3.4)
CHEMBL4078273	−8.1	12	5638	0.312	0.37	0.352	8	Glu86 (1.7) (2.9) (2.9) (2.0), Arg45 (2.4), Ser12 (2.3) (2.8), Thr16 (1.9)	1	Tyr46 (4.5)
Dihydrofolate reductase										
CHEMBL1966988	−8.7	11	6342	0.287	0.421	0.477	7	Ile37 (1.9) (2.6), Gly40 (2.2), Lys72 (2.5), Gly121 (1.8), Arg150 (1.8) (2.8)		
CHEMBL340488	−8.1	9	6330	0.311	0.353	0.343	7	Lys87 (1.9), Lys72 (2.1), Glu123 (2.8), Arg150 (2.9) (2.2) (2.9), Ser38 (2.2)		
ZINC000016682862	−8.0	11	7034	0.287	0.408	0.399	6	Arg150 (2.4), Ile37 (2.3), Gly121 (2.2), Tyr79 (2.8), Lys78 (2.1)	2	Phe52 (2.9), Tyr79 (2.7)
ZINC000828645375	−7.7	14	6246	0.351	0.518	0.521	8	Thr73 (2.1) (2.6), Gly121 (3.0) (2.2), Tyr125 (2.0), Ile37 (1.7), Ala29 (2.2), Tyr125 (2.0)	3	Met55 (3.0), Phe52 (2.7), Tyr79 (2.6)
CHEMBL3901573	−7.5	11	4999	0.306	0.411	0.369	8	Arg150 (2.5) (2.5) Ser38 (2.2), Gly121 (1.9), Ala199 (2.0), Tyr125 (2.4), Leu27 (2.1), Ala29 (2.1)	1	Met55 (3.8)
Methionine aminopeptidase 2										
CHEMBL3913373	−9	11	5428	0.299	0.311	0.371	9	Asn192 (2.1) (2.1), Ser206 (2.1), Glu226 (2.0) (2.9) (2.8), His194 (2.7), His202 (2.2), His244 (2.8)	1	His92 (2.6)
CHEMBL1962731	−8.9	15	5458	0.282	0.426	0.399	6	Asn192 (2.1), Glu226 (2.0), His202 (2.1), Asp113 (2.9) (2.4), Pro81 (2.6)	2	His202 (4.6), His92 (2.3)
CHEMBL3142997	−8.1	12	5264	0.323	0.436	0.536	8	Ser206 (1.8), Asn192 (1.8) (1.8) (2.2), Glu226 (2.6) (2.0), His202 (2.0) (2.3)		
ZINC000199197855	−8.0	13	5160	0.359	0.412	0.542	9	Ser206 (2.3), Gly193 (2.9), Asn192 (1.8) (1.9) (2.0) (2.0), Asp124 (2.8), Asp113 (2.1), His92 (2.4), His202 (2.9) (2.0)	2	His (4.5), Tyr304 (3.1)
ZINC000016682972	−7.9	15	5676	0.328	0.309	0.492	8	His202 (2.4), Glu226 (1.9), Gln317 (2.9), His92 (2.7), Asp113 (2.4) (1.9) (1.7)	2	His92 (4.7), Phe80 (5.0)

**Table 4 ijms-24-01412-t004:** The two-dimensional structures and binding energies of the best five compounds selected against the target proteins.

Molecular ID	Chemical Name	Chemical Structure	Binding Energy
Aquaporin			
CHEMBL3703838	(2S,3R,4R,5S,6R)-2-[4-chloro-3-[(4,4-dioxo-2,3-dihydro-1,4lambda6-benzoxathiin-6-yl)methyl]phenyl]-6-(hydroxymethyl)oxane-3,4,5-triol	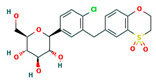	−36.0654 ± 2.6122
ZINC000002243083	2-Naphthol-3,6,8-trisulfonic acid	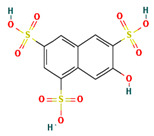	−2.8214 ± 4.9570
CHEMBL133039	Sodium;2-[3,5-dihydroxy-4-[3-(3-hydroxy-4-methoxyphenyl)propanimidoyl]phenoxy]acetate	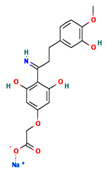	−30.5427 ± 3.1371
CHEMBL3140193	[(2S,3R,4S,5S,6R)-3,4,5-trihydroxy-6-(hydroxymethyl)oxan-2-yl] (1Z)-2,3-dichloro-N-sulfooxybenzenecarboximidothioate	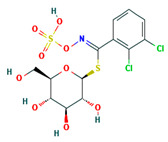	−21.3990 ± 6.5936
CHEMBL2132563	(2R,3R,4S,5R,6R)-2-[4-[3-(4-fluorophenyl)-1,2,4-oxadiazol-5-yl]-1,3-thiazol-2-yl]-6-(hydroxymethyl)oxane-3,4,5-triol	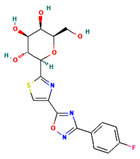	−32.1805 ± 4.0288
CTPsynthase			
CHEMBL48494	2-[(3-Amino-4,5,6-trihydroxyoxan-2-yl)oxymethyl]anthracene-9,10-dione	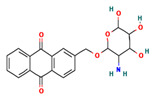	−37.1662 ± 3.5275
CHEMBL1162979	sodium;[(2S,3S,4R,5R,6R)-3,5-dihydroxy-2-[[3-hydroxy-5-[(Z)-2-(4-hydroxyphenyl)ethenyl]phenyl]methyl]-6-(hydroxymethyl)oxan-4-yl] sulfate	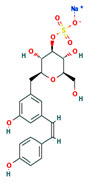	−45.4045 ± 2.6946
CHEMBL133039	Sodium;2-[3,5-dihydroxy-4-[3-(3-hydroxy-4-methoxyphenyl)propanimidoyl]phenoxy]acetate	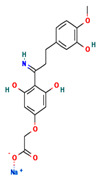	−38.1244 ± 2.4261
CHEMBL1091856	(5-{[4-(Benzenesulfonyl)-1,2,5-oxadiazol-3-yl]oxy}-1-hydroxy-1-phosphonopentyl)phosphonic acid	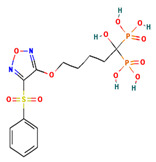	−56.5194 ± 5.0206
CHEMBL525202	(2S,3R,4R,5R,6R)-2-[[2-(benzylamino)-4-methyl-1,3-thiazol-5-yl]methyl]-6-(hydroxymethyl)oxane-3,4,5-triol	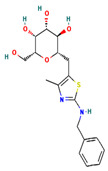	−44.3102 ± 6.8784
Thymidine kinase			
CHEMBL3674540	(E)-2-(Amino(1-(quinolin-6-ylmethyl)-1H-[1–3]triazolo[4,5-b]pyrazin-6-yl)methylene)hydrazinecarboxamide	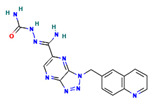	−25.6970 ± 4.4414
CHEMBL1683320	[(2R,3S,4R,5R)-5-[4-(4-fluorophenyl)triazol-1-yl]-3,4-dihydroxyoxolan-2-yl]methyl dihydrogen phosphate	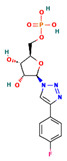	−26.6667 ± 4.9470
CHEMBL391279	(1S,3R,4R,7S)-3-(6-aminopurin-9-yl)-7-hydroxy-N-methyl-2-oxa-5-azabicyclo[2.2.1]heptane-1-carboxamide	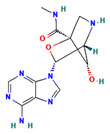	−9.1083 ± 4.8425
CHEMBL4078273	[(2R,3S,4S,5R,6S)-6-(4-hexylphenoxy)-3,4,5-trihydroxyoxan-2-yl]methyl hydrogen sulfate	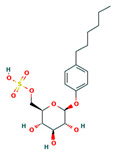	−33.5752 ± 4.6211
ZINC000031750813	3-(2,3-dihydro-1H-tetrazol-5-yl)-6-(2H-tetrazol-5-yl)-1,2,4,5-tetrazine	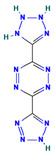	−20.8656 ± 3.4276
Dihydrofolate reductase			
CHEMBL1966988	2-[4-[(2Z)-2-(2-oxooxolan-3-ylidene)hydrazinyl]phenyl]sulfonylguanidine	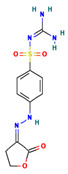	−28.0600 ± 3.0231
CHEMBL340488	((S)-2-Amino-propionyl)-sulfamic acid (2R,3S,4R,5R)-3,4-dihydroxy-5-(4-phenyl-thiazol-2-yl)-tetrahydro-furan-2-ylmethyl ester	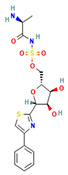	−35.9711 ± 3.1254
ZINC000016682862	ethyl 2-[[5-(benzoylcarbamothioylamino)-3,4-dicarbamoyl-1H-pyrrol-2-yl]sulfanyl]acetate	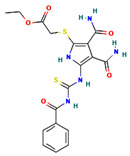	−23.2559 ± 3.125
ZINC000828645375	2-(4-chlorophenyl)-5-[methyl(2,3,4,5,6-pentahydroxyhexyl)amino]-1,3-oxazole-4-carbonitrile	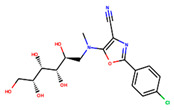	−28.9491 ± 6.23366
CHEMBL3901573	[(1R,2R,3S,4R)-2,3-dihydroxy-4-(pyridine-2-carbonylamino)cyclopentyl]methyl sulfamate	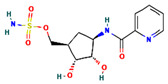	−13.5906 ± 6.5678
Methionine aminopeptidase 2			
CHEMBL3913373	1-(4-hydroxy-3-nitrophenyl)-3-[(2,4,6-trioxo-1-phenyl-1,3-diazinane-5-carbonyl)amino]urea	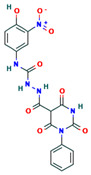	−43.5796 ± 5.1314
CHEMBL1962731	3-amino-5-(2-benzylsulfonylethylamino)-6-chloro-N-(diaminomethylidene)pyrazine-2-carboxamide	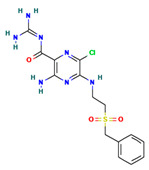	−29.9671 ± 9.8188
CHEMBL3142997	((5-(2,4-Dioxo-5-vinyl-3,4-dihydro-2H-pyrimidin-1-yl)-3-hydroxy-tetrahydro-furan-2-ylmethoxy)-hydroxy-phosphoryl)acetic acid	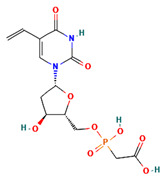	−30.0314 ± 5.2458
ZINC000199197855	[(3S,4R,5R)-3,4,5,6-tetrahydroxy-2-oxohexyl] (2S)-2-amino-3-phenylpropanoate	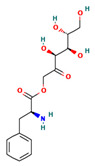	−28.6175 ± 6.0841
ZINC000016682972	ethyl (2S,3R,4R,5S)-5-(1,2,4-benzotriazin-3-yl)-2,3,4,5-tetrahydroxypentanoate	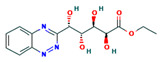	−32.6542 ± 6.1144

## Data Availability

All data needed to evaluate the conclusions in the paper are present in the paper and Appendix A.

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
