# Peer review of "Identification of Potential Druggable Targets and Structure-Based Virtual Screening for Drug-like Molecules against the Shrimp Pathogen Enterocytozoon hepatopenaei"

_ijms, 2023, doi:10.3390/ijms24021412_

Round 1

Reviewer 1 Report

This manuscript reports putative inhibitors for the treatment of slow growth syndrome caused by Enterocytozoon hepatopenaei (EHP) in shrimp. The authors conducted in silico screening against the five potential target proteins, aquaporin (AQP), cytidine triphosphate (CTP) synthase, thymidine kinase (TK), methionine aminopeptidase 2 (MetAP2), and dihydrofolate reductase (DHFR) with modeled structures by AlphaFold2. The five candidate compounds for each target were finally selected based on affinity, ADMET properties, electrostatic complementarity, complex stability, and binding energy. These compounds would contribute to the development of therapeutic agents for growth syndrome by EHP, if they are experimentally validated.

Comments:

1.     The reasons for target selection are unclear. Although the authors mentioned that they selected the targets based on the pathways and publications, how these proteins are suitable for the targets should be written based on the biological mechanism.

2.     The selectivity of the putative compounds for EHP proteins is unclear. The selected target proteins are also present in shrimp. Even if the sequence identities of EHP proteins with the host are low, the active sites are often conserved and similar. Therefore, the selected compounds could inhibit host proteins and exhibit toxicity. A detailed comparison of binding pockets and an analysis of binding modes will be needed.

Reviewer 2 Report

This is a systematic and comprehensive in silico study presented in a very well written manuscript. It would have been of greater interest if the findings were corroborated by experimental data; however, I understand that this is beyond the scope.

My recommendation is to add some background information in the discussion sections on the hit compounds. For example, what type of compounds are they? Are they drugs? Are they structurally related to which drugs? Is any activity known for them at other targets?

What definitely needs to be double-checked and corrected are the errors in the chemical structures in Table 3: The chemical structures for the second (ZINC000002243083) and third (CHEMBL133039) entries do not correspond to the given names. Please correct and check for correctness of the other entries.
